# Tomato Bushy Stunt Virus Nanoparticles as a Platform for Drug Delivery to Shh-Dependent Medulloblastoma

**DOI:** 10.3390/ijms221910523

**Published:** 2021-09-29

**Authors:** Chiara Lico, Barbara Tanno, Luca Marchetti, Flavia Novelli, Paola Giardullo, Caterina Arcangeli, Simonetta Pazzaglia, Maurizio S. Podda, Luca Santi, Roberta Bernini, Selene Baschieri, Mariateresa Mancuso

**Affiliations:** 1Laboratory of Biotechnology, Italian National Agency for New Technologies, Energy and Sustainable Economic Development, ENEA, Casaccia Research Center, Via Anguillarese 301, 00123 Rome, Italy; chiara.lico@enea.it; 2Laboratory of Biomedical Technologies, Italian National Agency for New Technologies, Energy and Sustainable Economic Development, ENEA, Casaccia Research Center, Via Anguillarese 301, 00123 Rome, Italy; barbara.tanno@enea.it (B.T.); luca.march46@gmail.com (L.M.); flavia.novelli@enea.it (F.N.); paola.giardullo@enea.it (P.G.); simonetta.pazzaglia@enea.it (S.P.); 3Laboratory of Health and Environment, Italian National Agency for New Technologies, Energy and Sustainable Economic Development, ENEA, Casaccia Research Center, Via Anguillarese 301, 00123 Rome, Italy; caterina.arcangeli@enea.it (C.A.); maurizio.podda@outlook.it (M.S.P.); 4Department of Agriculture and Forest Sciences (DAFNE), University of Tuscia, Via S. Camillo De Lellis, 01100 Viterbo, Italy; luca.santi@unitus.it (L.S.); roberta.bernini@unitus.it (R.B.)

**Keywords:** plant virus nanoparticles, tomato bushy stunt virus, medulloblastoma, targeting peptides, drug loading, molecular docking and simulation

## Abstract

Medulloblastoma (MB) is a primary central nervous system tumor affecting mainly young children. New strategies of drug delivery are urgent to treat MB and, in particular, the SHH-dependent subtype—the most common in infants—in whom radiotherapy is precluded due to the severe neurological side effects. Plant virus nanoparticles (NPs) represent an innovative solution for this challenge. Tomato bushy stunt virus (TBSV) was functionally characterized as a carrier for drug targeted delivery to a murine model of Shh-MB. The TBSV NPs surface was genetically engineered with peptides for brain cancer cell targeting, and the modified particles were produced on a large scale using *Nicotiana benthamiana* plants. Tests on primary cultures of Shh-MB cells allowed us to define the most efficient peptides able to induce specific uptake of TBSV. Immunofluorescence and molecular dynamics simulations supported the hypothesis that the specific targeting of the NPs was mediated by the interaction of the peptides with their natural partners and reinforced by the presentation in association with the virus. In vitro experiments demonstrated that the delivery of Doxorubicin through the chimeric TBSV allowed reducing the dose of the chemotherapeutic agent necessary to induce a significant decrease in tumor cells viability. Moreover, the systemic administration of TBSV NPs in MB symptomatic mice, independently of sex, confirmed the ability of the virus to reach the tumor in a specific manner. A significant advantage in the recognition of the target appeared when TBSV NPs were functionalized with the CooP peptide. Overall, these results open new perspectives for the use of TBSV as a vehicle for the targeted delivery of chemotherapeutics to MB in order to reduce early and late toxicity.

## 1. Introduction

Chemotherapy plays a pivotal role in the fight against tumors; however, systemic administration often results in uneven biodistribution and produces severe effects on healthy cells, limited accumulation in the target tissue and emergence of drug-resistant cancer cells [1]. To reach the central nervous system (CNS) and brain tumors with these drugs is even more challenging for the presence of the highly selective blood–brain barrier (BBB) [2]. Nonetheless, targeted treatments are particularly urgent for tumors such as Sonic Hedgehog-dependent medulloblastoma (SHH-MB) [3]. Arising in young children (<3 years of age), this MB variant has the highest risk of an unfavorable outcome [4].

In the last years, several nanoparticles (NPs)-based delivery systems have been developed in order to improve the pharmacokinetic and/or targeting of drugs [5]. These delivery systems, used to encapsulate, absorb or conjugate the therapeutic molecules, may allow the re-evaluation of compounds formerly considered too toxic for systemic administration.

Plant viruses, such as cowpea mosaic virus (CPMV) and tobacco mosaic virus, have been already considered for this kind of application. These viruses, noninfectious for animal cells, structurally stable, biocompatible and biodegradable, self-assembling nano-containers, easy to decorate with targeting ligands by genetic or chemical modification and rapidly produced at low costs using plants as biofactories, have proven to be excellent candidates for a wide range of applications [6].

We describe here the use of the tomato bushy stunt virus (TBSV) as novel candidate for the targeted delivery of drugs to murine brain tumors, Shh-MB in particular. TBSV, type member of the genus Tombusvirus, is characterized by an icosahedral capsid of ~ 32 nm in diameter made up of 180 subunits of the coat protein (CP). TBSV NPs can both encapsulate or expose on the surface small molecules and polypeptides [7] and are neither toxic nor teratogenic [8]. When intravenously injected into mice, these NPs do not induce alterations of tissues/organs, and, besides accumulating mainly in the reticuloendothelial system, they can be localized into the brain, also after perfusion [9].

Chimeric TBSV (cTBSV) NPs were designed and constructed to display peptides described in the literature to be endowed with the ability to target cancer cells and were produced on large scale in *Nicotiana benthamiana* plants. Once purified, MB targeting and uptake were tested in vitro in cultures of primary murine Shh-dependent MB cells derived from Patched1 heterozygous (*Ptch1^+/−^*) knockout mice, one of the most powerful and widely studied model of MB [10,11,12], and in vivo in the same animal model.

Overall, these results open new perspectives for the use of this TBSV-based delivery platform for the targeted drug delivery to MB, overcoming the BBB and reducing early and late toxicity.

## 2. Results

### 2.1. Construction, Production and Purification of WT and Chimeric TBSV NPs 

cTBSV NPs were constructed by engineering the viral genome to display on their surface tumor targeting peptides as fusion to the C-terminus of the CP. The peptide RGD, in single or double copy (RiGiD), found in the sequences of extracellular matrix proteins, was selected because it is able to bind to integrins, a family of transmembrane receptors overexpressed also on different types of tumor cells, and cTBSV carrying this peptide was used as a positive control [13]. The tLyp peptide is a linear truncated version of the cyclic peptide Lyp1, identified by the in vivo selection of a phage display peptide library, which through a C-terminal C-end rule motif (CendR) binds to neuropilin (NRP), overexpressed in angiogenic tumors, and activates cell internalization [14]. The CooP peptide is a glioblastoma homing peptide identified, such as Lyp-1 by the in vivo selection of a phage display peptide library, whose interacting partner has been identified primarily to be the mammary-derived growth inhibitor (MDGI/FABP3) [15]. Finally, the ApoE peptide is derived from Apolipoprotein E, one of the major proteins involved in lipid metabolism, which mediates the interaction with low-density lipoprotein receptor-related proteins overexpressed in the brain and in tumors herein [16] (Appendix A). A GGPGG linker was inserted between the CP and the peptide in order to avoid possible steric hindrance during CP folding and/or CP–CP interaction instrumental for virus particle assembly (Figure 1A–D), and at the same time to obtain a higher degree of flexibility, thus increasing the chance of proper interaction with cell surface receptors. A GPG spacer was also inserted between the two copies of RGD in the RiGiD chimera.

All the RNA transcripts obtained from the different viral constructs (TBSV-WT, -RGD, -RiGiD, -ApoE, -CooP, -tLyp1) were used to infect *N. benthamiana* plants and demonstrated the ability to induce on leaves the onset of typical infection symptoms (chlorotic vein clearing) (Figure 1E,F). Ten to eleven d.p.i. plant tissues were harvested, RNA extracted, retrotranscribed and cDNA fragments sequenced, confirming the presence of the heterologous sequences. Total protein extracts, containing cTBSV, were also obtained and used to infect a second set of plants, allowing us to verify the genetic stability of the chimeric viruses through subsequent infection cycles. On this basis, all the constructs were then used to produce the cTBSV on large scale. Particles were purified with an average yield of 1 mg/g of fresh leaves tissue weight. No substantial differences in recovery were observed among WT and cTBSV, indicating that the genetic modifications did not affect viral fitness and in planta behavior. Each batch of purified virus particles analyzed by Coomassie Blue staining after SDS-PAGE confirmed the presence in all samples of the viral CP (both as a monomer or as dimer/aggregate) and purity of each preparation (Figure 1G).

### 2.2. In Vitro Validation of cTBSV Uptake

Shh-MBs cells were isolated from *Ptch1^+/−^* mice and maintained in culture in the appropriate medium. Subsequently, the cells were incubated for 3 h with cTBSV, or TBSV-WT as control, to be then collected and used to extract RNA and verify the presence of the viral genome by qRT-PCR. All cTBSV were internalized by MB cells more efficiently than the WT virus. Data based on qPCR analysis revealed that the uptake of TBSV-RGD, -RiGiD, -ApoE, -tLyp1 and -CooP was 69, 37, 40, 99 or 123 times the uptake of TBSV-WT, respectively (Figure 2A). Moreover, referring the internalization of the cTBSV NPs to TBSV-RGD (a well characterized tumor-targeting peptide) [13], it was observed that the uptake of TBSV-tLyp1 and -CooP was increased by 1.43 and 1.78 times, respectively, while the internalization efficiency of TBSV-RiGiD and -ApoE was reduced (0.54 and 0.59 times respectively). On this basis, the subsequent functional characterization studies were carried out only on TBSV-tLyp1 and TBSV-CooP.

GCPs are the highly proliferating and undifferentiated cells residing in the external granular layer of the cerebellum during its postnatal development, from which MB arises [17]. After isolation and purification from *Ptch1^+/−^* mouse cerebella at P2, GCPs were incubated with TBSV-WT, -RGD, -tLyp1 or -CooP. The results of qPCR analysis revealed that, compared to TBSV-WT, the internalization of TBSV-tLyp1 and -CooP was 8.2- and 4.7-fold higher, respectively, while the uptake of TBSV-RGD was a 0.46-fold lower (Figure 2B, left panel). These experiments were repeated on GCPs induced to differentiate in order to mime the physiological process occurring during the cerebellum development (Figure 2B, right panel). Results showed that internalization of TBSV-tLyp1 and TBSV-CooP was significantly reduced (2.21- and 4.69-folds lower, respectively) compared to the uptake by undifferentiated GPCs, although TBSV-tLyp1 was still internalized more efficiently than the WT virus.

### 2.3. Immunolocalization of the Interaction Partners

It has been established that the interacting partners of CooP and tLyp1 peptides are the mammary-derived growth inhibitor (MDGI/H-FABP/FABP3) [18] and neuropilin 1 (NRP-1) [19], respectively. To evaluate if the specific uptake of the cTBSV NPs displaying these peptides may be mediated by these proteins, their expression and localization in Shh-MB cells, GCPs and differentiated GCPs were evaluated by immunofluorescence. As shown in Figure 2C, MB cells and GCPs showed a high expression of both NRP-1 and FABP3, which conversely are not detected in differentiated GCPs. Altogether these results suggested that the specificity of the uptake of TBSV-tLyp1 and TBSV-CooP by MB cells and GCPs may be receptor-mediated.

### 2.4. Docking and MD Simulation of the tLyp1-NRP-1 and CooP-FABP3 Complexes 

The possible binding mode of the tLyp1 and CooP peptides (both in their free version or as fusion to the viral CP) with their respective interacting receptors was investigated by molecular docking and MD approaches. MD simulations were employed to enlighten the structural changes occurring in the active binding sites and to monitor the overall stability of the complexes predicted by docking calculations (Appendix A). As reference, the interaction mode of tuftsin with NRP-1 and of oleic acid with FABP3, natural binders of these receptors, were also simulated. The clustering of the structural conformations adopted by the different complexes during the MD simulation showed that the natural ligands form complexes with their receptors for approximatively 98% of their trajectory (panels A and B of Appendix A) and that such stability is mainly due to the formation of persistent H-bonds (panels C of Appendix A).

Concerning the tLyp1-NRP-1 complex, as shown in Figure 3A, approximatively 67% of the trajectory was mainly described by two of the 21 identified structural clusters. After 72 ns of simulation, the MD trajectory indicated that several transitions among the clusters occurred, suggesting that the complex was unstable. Although the hydrogen-bonding pattern (Appendix A) appeared rich in interactions between tLyp1 and NRP-1, only two H-bonds (Gly318-Arg7 and Asp320-Arg5) showed occupancy for more than 40% of the trajectory. After about 72 ns of the simulation, the tLyp1 peptide moved out from the receptor active site and the contact was maintained by the interaction of Arg7 with Glu319 of NRP-1 (Figure 3B). Interestingly, when tLyp1 is fused to the CP (Figure 3C), the binding to NRP-1 appeared to be more stable, with several H-bonds (Appendix A) contributing to stabilization. After about 60 ns of simulation, some residues (Gly375 and Lys397) of the distal part of NRP-1 appear to interact also with Asp191 of the viral CP, giving a further contribution to the attraction towards the cTBSV. The time evolution of the binding free energy (Appendix A) shows that after 75 ns of simulation, the interaction of free tLyp1 with the receptor becomes unstable (∆G = −43.229 kJ/mol), whereas the conjugation of tLyp1 to the viral CP seems to enhance of two times the strength of the binding with NRP-1 (∆G = −91.07 kJ/mol).

The dynamics of the interaction of CooP with FABP3 (Figure 4) seemed to be very stable and comparable with that of oleic acid (Appendix A). In particular, the interaction of Ala9 of CooP with Arg126 in the active site of FABP3 was maintained all over the simulation time (Appendix A).

The binding to FABP3 of the CooP peptide fused to the TBSV CP seems to be stronger compared to the free peptide despite few changes in the H-bonding pattern (Appendix A). The values of the binding free energy (Appendix A) clearly show that CooP peptide fused to the viral CP recognize with almost 3-fold higher binding affinity the FABP3 receptor (∆G = −377.07 kJ/mol vs. ∆G = −129.30 kJ/mol).

These results, by supporting the idea that internalization is receptor-mediated, indicate also that the interaction of the peptides with their receptors, CooP in particular, is strengthened by fusion to the viral CP.

### 2.5. TBSV-tLyp1 and TBSV-CooP Characterization and Loading with DOX

The size homogeneity and monodispersion of TBSV-WT, -tLyp1 and -CooP preparations were verified by TEM and dynamic light scattering (DLS), before proceeding with DOX-loading. Measurement of the NPs on TEM micrographs rendered a mean size of 32 nm (Table 1), consistent with the expected dimensions of TBSV. On the other hand, DLS analysis indicated that the mean diameter of the NPs was higher, suggesting the formation of agglomerates. This hypothesis is supported by the analysis of the poly-dispersity index that, even if below 0.7 as required by ISO standard ISO 22,412:2017 to define a polydisperse distribution of particles ranged from 0.419 to 0.588 (Table 1). The presence of aggregates was detected in particular in the TBSV-CooP and TBSV-tLyp1 samples and this may be due to the fact that the ζ-potential values of these NPs is closer to neutrality, thus less prone to repulsion (Table 1).

In order to “load” TBSV NPs with DOX, an ad hoc protocol was developed. Positively charged DOX was added in 5000 molar excesses to the virus particles in a swelling buffer in the presence of EDTA, in order to facilitate access into the viral cavity and interaction with negatively charged genomic RNA. Then, by restoring the initial pH conditions and proper Ca^2+^ and Mg^2+^ concentrations, the viral capsid was induced to re-establish the compact state, entrapping the drug into the shell. DOX in excess was removed from the samples performing a sucrose cushion passage. Electrophoresis analysis confirmed that DOX migrates together with TBSV (Figure 5) and not towards the negative pole as it would be the case when it is free. All the TBSV NPs preparations showed a reproducible loading capacity in terms of ng of DOX/µg of TBSV: 100 for TBSV-WT, 113 for TBSV-RGD, 150 for TBSV-tLyp1 and 157 for TBSV-CooP, corresponding to 1543, 1736, 2314 and 2424 DOX molecules/virion, respectively. Results in Table 1 show that the encapsulation efficacy (EE) ranged from 31 to 48% and the loading capacity (LC) from 10 to 16%.

### 2.6. cTBSV-Mediated Delivery of DOX to Shh-MB Cells

The ability of the cTBSV to deliver DOX was evaluated in MB cell cultures, measuring the cell viability after 72 h of incubation with different concentrations of free-DOX or DOX-loaded NPs (TBSV-WT, -RGD, -tLyp1 or -CooP). The viability of the cells treated with the unloaded WT virus was set as 100%. 

As shown in Figure 6A, cell viability was significantly reduced by free-DOX in a dose-dependent manner, with maximum efficacy reached at a concentration of 25 µM (90% of dead cells). DOX delivery through TBSV-WT and -RGD was able to significantly increase cell killing at lower DOX concentrations (around 85% of cell-death with TBSV-WT and 75% with TBSV-RGD at 10 µM vs. 50% with free-DOX). Moreover, the largest difference in cell-death rate among groups can be appreciated at the lowest DOX concentration (5 µM), when delivered through TBSV-tLyp1 and TBSV-CooP producing over 90% of cell-death rate compared to 68% in TBSV-RGD, 46% in TBSV-WT and only 17% in free-DOX groups, indicative of the potential therapeutic gain of this targeted strategy of delivery.

### 2.7. cTBSV NPs Targeting to Shh-MB In Vivo

To verify that TBSV-tLyp1 and TBSV-CooP were able to specifically target MB also in vivo, *Ptch1^+/−^* mice developing the tumor were IV injected with TBSV-WT or chimeric particles and 24 h later the absolute quantity of TBSV NPs in MB and NB was determined by qPCR. The analysis (Figure 6B) indicated that TBSV-WT, as well as chimeric NPs, were able to reach the brain. Notably, even if the overall amount of TBSV-WT was higher compared to that of cTBSV NPs, TBSV-CooP showed the highest specificity in targeting MB cells (3.56-fold increase vs. NB, *p* < 0.0001) with respect to TBSV-WT and TBSV-tLyp1 (2.47- and 2.51-fold, respectively).

## 3. Discussion

In the present study, NPs based on TBSV have been developed with the aim of targeting MB, a primary CNS tumor, which is one of the most frequent malignancies in childhood. Rapid advances in molecular genetics over the past two decades have provided consensus on the existence of four distinct molecular MB subgroups: Wingless (WNT), Sonic Hedgehog (SHH), group 3 and group 4 [3]. Accordingly, a new concept of therapeutic approach is now evolving, which could substitute or at least complement the standard treatment of MB, currently based on resection, chemotherapy and craniospinal irradiation. This approach is particularly relevant for SHH-MBs, the most common subgroup in infants (≤3 years old) in whom radiotherapy is precluded due to the severe neurological side effects. Targeted therapy is an example of alternative treatment and Shh pathway antagonists, primarily those inhibiting smoothened (SMO), are currently in clinical trials demonstrating improved progression-free survival [20]. However, such inhibitors might be ineffective when administered as monotherapy, due to the primary resistance observed in patients, or to the secondary resistance related to mutations occurring in the targeted receptor or in other involved proteins [21]. More efforts are thus necessary to develop new therapeutic options directed against Shh-MB and smart drug delivery systems represents the most promising. A wide range of different NP-based delivery systems have been investigated pre-clinically, which may lead to more NPs reaching the clinic. Glioblastoma (GBM) is the most explored tumor brain to test the efficacy of these type of devices, and 10 different nanomaterials are currently evaluated for possible clinical use [22]. Different NPs have been experimentally tested to investigate efficacy also against MB, including poly(lactic-co-glycolic acid) conjugated to polyethylene glycol (PLGA-PEG)-based NPs delivering the Shh pathway inhibitor HPI-1 [23,24], high-density lipoprotein (HDL) NPs [25], biomimetic NPs decorated with a ligand targeting the stage-specific embryonic antigen 1 (SSEA-1) (anti-CD15) and loaded with the Shh inhibitor sonidegib (LDE225) [26], or polyoxazoline block copolymer micelles encapsulating the Shh pathway inhibitor [27].

In recent years the interest is increasing towards the use of biomaterials for nanomedical applications. Building blocks of biological origin are indeed ideal to construct biocompatible nanodevices for drug delivery. Viruses, and their non-infectious structural analogues (i.e., virus-like particles), are bio-based nanoparticles. The self-assembling properties of viral capsid proteins ensure the spontaneous formation of shells that are structurally uniform at the atomic level. In addition, targeting to specific cellular types may be in some cases intrinsic because related to viral tropism. In this context, plant viruses may be considered as a “natural collection” of NPs with conformations that range from spherical, to tubular or filamentous. Not being infectious for animal cells, they are replication incompetent thus intrinsically safe, and their half-life in vivo cannot be hindered by pre-existing immunity. These NPs may be rapidly produced on large scale at low costs using plants, in particular *N. benthamiana* that is permissive to the infection by a wide range of different plant viruses [28]. The intrinsic ability of plant viruses to target specific mammalian cells is not obvious; however, CPMV spontaneously binds to cervical and colon cancer cells through vimentin [29], a cell marker expressed also on GBM [30]. Nonetheless, specific targeting may be easily obtained decorating the viral CP with “homing” peptides by viral genome engineering or chemical interventions.

By these approaches, a plethora of plant viruses, some of which, including TBSV, characterized for their biodistribution and formally demonstrated to be devoid of toxicity, have been investigated for very different applications in nanomedicine, ranging from vaccine formulations to tissue engineering, imaging and diagnostic. Concerning drug delivery, the potentialities of plant viruses as possible vehicles have been tested mainly on melanoma, myeloma, colon, breast, prostate and ovarian cancer, both in vitro and in vivo, and only rarely on brain tumors [31]. In this scenario, the development of new plant virus-based delivery platforms appears to be important to increase the number of well characterized tools available for applications in biomedicine.

Here we focused on TBSV NPs, displaying on the outer surface peptides selected from a larger panel whose structural, dynamical and physicochemical features were previously characterized in detail by exploiting the capabilities of MD simulation [32]. All the peptides were correctly displayed and were able to induce a significant increase in the uptake of TBSV by primary Shh-MB cells. In particular, CooP and tLyp1 chimeras mediated a considerably higher internalization. To explain the advantage offered by these two peptides, the binding mode of tLyp1 and CooP (both in their free version or as fusion to the viral CP) with their interacting partners, NRP-1 and FABP3, respectively, were investigated by MD approaches. Interestingly, the dynamics of the interaction resulted to be more stable when peptides, CooP in particular, were fused to the CP, supporting the idea that internalization in MB cells is indeed receptor-mediated. NRP-1, involved in cell survival and proliferation, seems to favor an undifferentiated phenotype in cancer cells [33] and its overexpression has been reported to be correlated with a poor prognosis in both GBM [34] and MB [35]. Furthermore, it has been demonstrated that 12-aminoacids within the NRP1 cytoplasmic domain are necessary to regulate Hedgehog signaling [36]. In agreement, we revealed NRP1 expression only in MB cells and in GCPs, both strictly depending on Shh deregulation. A similar trend has characterized the immunostaining of FABP3, although, to the best of our knowledge, no correlation with Shh-dependence is reported in the literature. However, FABPs are expressed in the developing brain and their cellular distribution and genetic variation have been implicated in a number of brain diseases including human brain cancer. In particular, in a meta-analysis study of gene expression in murine and human MB, FABP3 has been reported to be downregulated [37]. The absence of expression of both receptors in differentiated cells well correlates with the significant reduction of cTBSV internalization, corroborating the high specificity for MB targeting offered by these NPs, with strong implications toward the safety of differentiated tissues. The functional test carried out in vitro with DOX-loaded cTBSV NPs highlighted that all NPs offered a significant advantage in terms of efficacy with respect to the delivery of low concentrations of free DOX. Five µm DOX delivered through TBSV-tLyp1 and TBSV-CooP were sufficient to induce a mortality rate of MB cells of 90%, a dose five-fold lower than that necessary to induce the same mortality using the free drug. Importantly, at the same DOX-loaded concentration, these cTBSV particles offered a statistically significant reduction of tumor cell viability compared to TBSV-RGD and TBSV-WT. Overall, these results are in agreement with data showing an increase in the specific uptake of both chimeras by MB cells and GCPs. Notably, when tested in vivo, CooP peptide increased the uptake of the plant virus NPs by MB cells compared to that of the unmodified NPs and to cTBSV displaying the peptide tLyP1 (3.5-fold compared to around 2.5 in the other cases). Beside the same experiment showed that the access to the brain of the cTBSV particles is overall less efficient if compared to that of the WT particles. This may be due at least to three main factors i.e., the slight tendency to aggregation of the chimeric particles when suspended in saline and/or the delivery route and/or deposition of a different protein corona. Moreover, it has to be considered that the cellular composition of the tumor mass is less homogenous if compared to that of the cells in the in vitro cultures.

## 4. Materials and Methods

### 4.1. TBSV Genetic Engineering, Production in Plants, Purification and Characterization

In order to construct the cTBSV NPs, the sequences encoding tumor targeting peptides were inserted in the TBSV–vector at the 3’-end of the p41 gene, encoding the C-terminus of the CP [7]. The vector was digested ApaI/PacI (New England Biolabs, Ipswich, MA, USA) and ligated with DNA fragments with compatible protruding ends obtained by the in vitro annealing of synthetic oligonucleotides couples encoding the peptides (Appendix A). The oligonucleotides were designed using preferential *N. benthamiana* codon usage. To obtain infectious RNA transcripts, all the vectors (pTBSV, pTBSV-RGD, pTBSV-RiGiD, pTBSV-ApoE, pTBSV-CooP and pTBSV-tLyp1) were linearized by XmaI digestion and in vitro transcribed using the MEGAscript T7 High Yield Transcription kit (Thermo Fischer Scientific, MI, Italy). Each RNA was used to inoculate six to eight weeks old *N. benthamiana* plants, grown under controlled conditions (24 °C, 16 h light/8 h dark) in a containment greenhouse, by abrading the adaxial side of 2 leaves/plant with carborundum (silicon carbide; VWR International, Radnor, PA, USA). Two cycles of reinfections were performed using saps from infected leaves prepared by homogenizing the systemic tissue of an infected plant in 1xPBS. After each cycle of infection, the onset of symptoms was monitored, the RNA was extracted from symptomatic systemic leaves with the RNeasy plant mini kit (QIAGEN, Hilden, Germany) following manufacturer’s instructions, and the RT-PCR was performed using the QuantiTect Reverse Transcription Kit (QIAGEN, Hilden, Germany), with specific primers up- and downstream the TBSV cp gene 3’ end (Appendix A). Finally, the obtained PCR fragment was analyzed by sequencing (BMR Genomics, Padova, Italy). TBSV particles were purified following a previously developed protocol [7]. Briefly, infected leaves were collected and ground to a fine powder. After plant material homogenization with 3 mL/g of 50 mM sodium acetate, pH 5.3, supplemented with 1% ascorbic acid and a cocktail of protease inhibitors (Sigma, Saint Louis, MO, USA), the extract was clarified by low-speed centrifugation (8000× *g* for 15 min at 4 °C). After adjusting the pH to 5.3, the supernatant was ultra-centrifuged for 1 h at 90,000× *g* at 4 °C. The obtained pellet was finally resuspended in 50 mM sodium acetate, pH 5.3.

### 4.2. Mouse Model 

A colony of mice lacking one *Ptch1* allele (*Ptch1^+/–^*), derived by gene targeting of 129/Sv ES cells and maintained on the CD1 strain background, is continuously maintained in the ENEA Casaccia (Rome, Italy) animal facility by crossing *Ptch1^+/−^* heterozygous males with CD1-wild-type females and vice versa, and genotyped as previously described [12]. Animals are housed under conventional conditions with food and water available ad libitum and a 12 h light–dark cycle, thus observed daily upon unequivocal appearance of full-blown symptomatic MB (severe weight loss, paralysis, ruffling of fur, inactivity).

### 4.3. Primary Cell Cultures and TBSV NPs Internalization

After tumor dissection, dissociated cells were maintained in complete medium UltraCULTURE™ cell culturing medium (Lonza BioWhittaker Inc., Basel, Switzerland) supplemented with 2 mM L-glutamine (Lonza BioWhittaker Inc., Basel, Switzerland), 1% Penicillin-Streptomycin (Life Technologies, Carlsbad, CA, USA), SAG dihydrochloride (Sigma Aldrich, Saint Louis, MO, USA) and 20% fetal bovine serum (Euroclone, Milan, Italy).

Granule Cell Precursors (GCPs) were purified from *Ptch1^+/–^* mouse cerebella at postnatal day 2 (P2) as previously described [38] and maintained in culture in DMEM/F12 (Gibco, Amarillo, TX, USA) supplemented with 0.6% glucose (Sigma Aldrich, Saint Louis, MO, USA), 25 μg/mL insulin (Sigma Aldrich, Saint Louis, MO, USA), 60 μg/mL N-acetyl-l-cystein (Sigma Aldrich, Saint Louis, MO, USA), 2 μg/mL heparin (Sigma Aldrich, Saint Louis, MO, USA), 20 ng/mL EGF (PeproTech, Inc., East Windsor, NJ, USA), 20 ng/mL bFGF (PeproTech, Inc., East Windsor, NJ, USA), 1X penicillin-streptomycin and B27 supplement without vitamin A (Gibco, Amarillo, TX, USA). To induce differentiation, GCPs were seeded in DMEM/F12 with N2 supplement (Gibco, Amarillo, TX, USA) and 2 μg/mL heparin, 0.6% glucose, 60 μg/mL N-acetyl-l-cysteine, containing 1% Calf Serum (Euroclone, Milan, Italy) and PDGF 10 ng/mL (PeproTech, Inc., East Windsor, NJ, USA) for 7 days.

MB cells were seeded at the density of 50,000 cells/cm^2^. Twenty-four hours later, they were incubated with the NPs at the concentration of 6.6 ng/µL for 3 h. GCPs and differentiated GCPs were seeded at the density of 20,000–50,000 cells/cm^2^, respectively. Seven days later, both cell types were incubated with the NPs for 3 h. To analyze the uptake of the functionalized TBSV by different cell lines, RNA isolation and qPCR were performed as below described.

### 4.4. RNA Isolation and Real-Time qPCR (qPCR) Analysis

RNA isolation from cells was performed with the RNeasy Mini Kit (QIAGEN, Hilden, Germany). cDNA synthesis was performed using the High Capacity cDNA reverse transcription kit (Applied Biosystems, Foster City, CA, USA), and qPCR was carried out by StepOnePlus™ Real-Time PCR System (Applied Biosystems, Waltham, MA, USA), using Power SYBR^®^ Green PCR Master Mix (Applied Biosystems, Waltham, MA, USA). Relative gene expression was quantified using Glyceraldehyde-3-phosphate (*Gadph*) as house-keeping gene. Oligonucleotide primers used for quantitative RT-PCR are listed in Appendix A. The ΔΔCt quantitative method was used to normalize the expression of the reference gene and to calculate the relative expression level of the TBSV cp target gene.

### 4.5. Immunofluorescence Analysis

To detect FABP3 and NRP-1, MBs, GCPs and differentiated GCPs cells were seeded on Lab-Tek Flask previously treated with Poly-D-lysine (Sigma-Aldrich, Saint Louis, MO, USA) and allowed to adhere for 16 h. Cells were fixed with 4% paraformaldehyde for 15 min at RT, permeabilized with 1xPBS, 0.1% triton X-100 for 2 min on ice and incubated in 1xPBS 5% bovine serum albumin (BSA). Cells were incubated overnight with anti-NRP-1 (monoclonal 1:300; Abcam, Cambridge, UK) and anti-FABP3 (H-FABP; polyclonal 1:100; Abcam, Cambridge, UK), and subsequently with the secondary antibody Alexa Fluor 488 (1:1000; Invitrogen) and DAPI (1:1000; Thermo Fisher, Waltham, MA, USA). Images were acquired with an Eclipse 80i Fluorescence Microscope (Nikon, Tokyo, Japan), equipped with the Imaging Software NIS-Elements BR Version 3.2.

### 4.6. Molecular Docking and Molecular Dynamics Simulations 

The structures of human NRP-1 (PDB id: 5IYY) [39] and FABP3/H-FABP (PDB id: 5CE) [40] were retrieved from RCSB Protein Data Bank and used as starting co-ordinates for the docking calculations and simulations. The structures of tLyp1 (CGNKRTR) and CooP (CGLSGLGVA) peptides were taken from our previous work [32]. Modeler [41] was used for generating the structural models of the asymmetric unit of the cTBSV in which, at the C-terminus of each CP, the tLyp1 or the CooP peptides were inserted with a GGPGG linker. Details on model building are provided in Appendix A. ClusPro protein–protein docking [42,43,44] and HADDOCK 2.2 [45] were used to obtain the receptor-peptide and receptor-ligands complexes, respectively. As positive controls, tuftsin (TKPR, a peptide known to bind NRP-1) and oleic acid (CH3(CH2)7CH=CH(CH2)7COOH, C18H34O2, the natural ligand of FABP3/H-FABP) were docked with their respective targets. In both docking methods, standard protocols were used and an ensemble of ten poses was generated for each docking run. The receptor-peptide/ligand complexes with the lowest binding energy were selected from the ensemble and used as the starting configuration for the molecular dynamics (MD) simulation as summarized in Appendix A. All the MD simulations were performed using Gromacs 2019 [46] and the trajectories were visualized and analyzed with VMD [47] with the tools included in Gromacs. Details on the MD procedure and methods of analysis are reported in the Appendix A.

### 4.7. Transmission Electron Microscopy

NPs morphology was characterized by transmission electron microscopy (TEM). For TEM images, samples were fixed using 4% paraformaldehyde (PFA) in 1xPBS, then placed on formvar-carbon coated copper grids, washed and negatively stained with 2% uranyl acetate. Samples were observed at a JEOL 1200 EX II electron microscope, and images were acquired by the Olympus SIS VELETA CCD camera equipped with the iTEM software. To calculate virions diameter, micrographs were analyzed using scientific image manipulation software ImageJ (National Institutes of Health, USA). TEM analysis was performed by the Electron Microscopy Lab, Centro Grandi Attrezzature, (CGA) University of Tuscia, Viterbo.

### 4.8. Dynamic Light Scattering and ζ-Potential Analyses 

Particles size, ζ-potential and the polydispersity index of NPs were determined by dynamic light scattering (DLS) and electrophoretic light scattering (ELS) using a Zetasizer Nano ZS instrument (Malvern-Panalytical, Malvern, UK) and the Malvern Zetasizer software for data analysis. The ζ-potential was calculated using the mixed mode measurement phase analysis light scattering (M3-PALS) technique (Malvern-Panalytical, Malvern, UK). Twenty µL of a 2 µg/µL viral nanoparticles solution were diluted in a Folded Capillary Zeta Cell (for ELS) and in a 12 mm Square Polystyrene Cuvettes (for DLS) to a final volume of 800 µL using a physiological solution (0.9% NaCl). Data were collected at 25 °C for 120 s (for DLS) and 180 s (for ELS) in triplicate. The data reported represent the size by intensity, as diameter of the peaks. All the analyses were performed by IXTAL srl, Novara, Italy.

### 4.9. Production of DOX Loaded TBSV NPs

Approximately 2 mg/mL of purified TBSV NPs were incubated in swelling buffer (0.1 M Trizma base, 50 mM EDTA, pH 8.0) for 1 h at RT in agitation. A 5000 molar excess of DOX (0.324 µg DOX/µg virus) was then added and incubated overnight at 4 °C on a bench-top tube rotator. Samples were then incubated in an association buffer (0.2 M Na Acetate, 25 mM CaCl2, 25 mM MgCl2, pH 5.2) for 1 h in agitation at RT to reassemble the virus cages. DOX excess was removed by overlaying the preparation onto a 20% sucrose cushion ultracentrifuged at 100,000× *g* for 1 h at 4 °C in a swing-out rotor. Pellets were resuspended overnight in association buffer and stored at 4 °C. To confirm the association of DOX to TBSV, about 10 µg of the NPs were separated on a native 0.8% TBE agarose gel and visualized under UV and white light before and after Coomassie Blue staining. Finally, TBSV-DOX NPs were analyzed by a UV/VIS spectrophotometer (Shimadzu UV-2600) at 485 nm to determine the number of DOX molecules loaded/virion. A DOX calibration curve was generated, and the Lambert–Beer law used to determine the concentration of DOX associated to NPs [48]. The drug encapsulation efficiency (EE) and loading capacity (LC) of DOX in NPs were also determined according to the formula:

EE(%) = Amount of DOX in the NPs/total amount of DOX added (×100%) 

LC(%) = Amount of DOX in the NPs/NPs weight (×100%) 

### 4.10. Cell Viability Assay

Cell viability was measured using the RealTime-Glo™ MT Cell Viability Assay (Promega, Milan, Italy). MB cells (5000 cells/well) were plated in a 96-well plate according to the manufacturer’s instructions. DOX-cTBSV concentrations were adjusted to maintain constant DOX concentration (ranging from 5 µM to 25 µM) and luminescence was measured every 24 h using the plate-reader GloMax (Promega, Milan, Italy) up to 72 h.

### 4.11. Absolute Quantification of c-TBSV in Target Tissues after Intravenous Injection

Mice with clear MB symptoms (4 females and 5 males) were injected with an IV with 200 µg of purified TBSV-CooP, TBSV-tLyp1 or TBSV-WT. Twenty-four hours later the animals were euthanized, brains explanted and MB isolated from normally appearing brain tissue (NB). MB and NB were weighted, and cells counted after proteolytic dissociation with Accutase (Euroclone, Milan, Italy). Total RNA was extracted from approximately 30 mg of MB or NB homogenized with gentleMACS™ Octo Dissociator (Miltenyi Biotec, Bergisch Gladbach, Germany) and RNA extracted as described in Appendix A.

Total RNA (1 µg) was reverse transcribed and 1/30 used to quantify the number of viral RNA genome copies through qPCR. Reactions were performed in triplicate from each biological replicate and normalization based on 5 housekeeping genes, i.e., *Gadph*, *β**-Actin*, *TATA box binding protein* (*TBP*), *ribosomal protein L13A* and *32* (*RPL13a* and *RPL32*). Oligonucleotides used are summarized in Appendix A. The absolute quantification of the viral RNA in each sample was performed by means of absolute quantification method using a standard curve based on TBSV RNA. Knowing the molecular mass of the TBSV genome, the obtained value was converted in the number of copies of viral genome (i.e., number of particles) and then in the number of viral particles/1 × 10^6^ cells. This was possible on the basis of the knowledge generated by a preliminary experiment in which MB and NB were collected from 16 *Ptch1^+/–^* mice and used to count cells and to extract total RNA. These preliminary data indicated that 1 mg of MB is made by a mean of 434.888 ± 33.058 cells and yields 1.188 ± 0.192 μg of RNA, while 1 mg of NB tissue is made by a mean of 149.937 ± 4.409 cells and yields 0.258 ± 0.046 μg RNA.

### 4.12. Statistical Analysis

All quantitative data were presented as mean ± SD and statistical significance (p) was calculated by a two-tailed Student’s *t*-test. All analyses were carried out using Graphpad Prism 6 Software.

## 5. Conclusions

In summary, the presented data support the hypothesis that cTBSV NPs, and TBSV-CooP in particular, might be suitable bio-based vehicles, rapidly produced on a large scale at low costs using plants, for the targeted delivery of chemotherapeutics to Shh-dependent MB, potentially overcoming problems related to low drug bioavailability and adverse off-target effects.

## Figures and Tables

**Figure 1 ijms-22-10523-f001:**
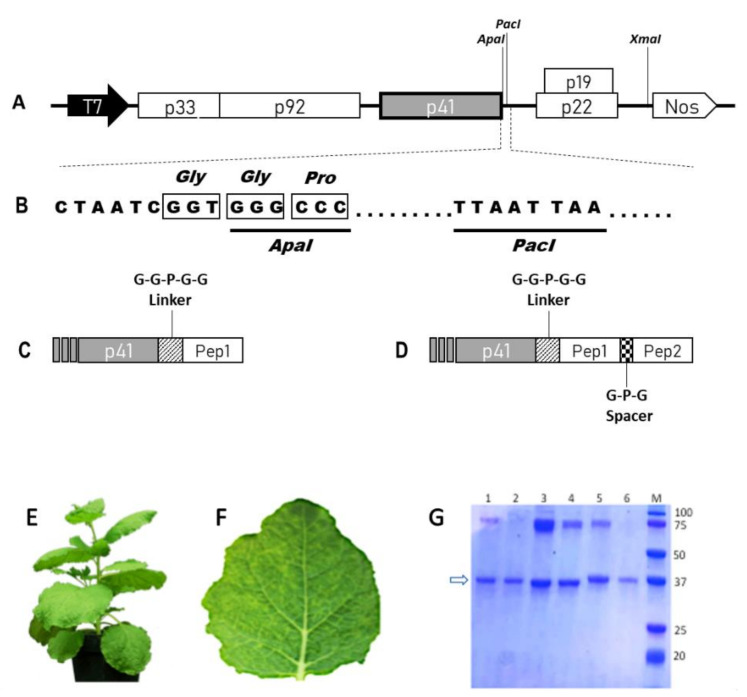
Schematic representation of TBSV cDNA constructs and TBSV NPs characterization. (**A**) Viral genome organization in the TBSV–vector plasmid. The 5 ORFs and the respective encoded proteins with their molecular masses (in kDa) are represented as boxes: p33 and p92 (RNA-dependent RNA polymerase), p41 (CP), p22 (movement protein), p19 (silencing inhibitor). T7—promoter sequence from T7 phage; Nos—terminator sequence from the Agrobacterium tumefaciens nopaline synthase gene. The XmaI restriction site used for vector linearization is also indicated. (**B**) Sequence detail of the cp 3’ region in the TBSV–vector. The underlined sequences represent the restriction enzyme sites used to clone the heterologous sequences. (**C**,**D**) Schematic representation of single or double peptides coding sequences fused to the viral cp. As simplification, only the 3’ region of the cp gene is illustrated. Striped box—GGPGG linker coding sequence; dotted box—GPG spacer coding sequence. (**E**) Adult *N. benthamiana* plant. (**F**) Typical TBSV symptoms (mainly chlorotic spots and vein clearing) induced at 7 d.p.i. (**G**) Coomassie Blue stained SDS-PAGE of TBSV NPs (5 µg/lane) purified from infected *N. benthamiana* plants. M: Precision Plus Protein Dual Xtra Prestained Protein Standard (Bio-Rad); (1): TBSV-WT; (2): TBSV-RGD; (3): TBSV-CooP; (4): TBSV-tLyp1; (5): TBSV-ApoE; (6): TBSV-RiGiD. The molecular masses of the marker bands are indicated. The arrow indicates the CP.

**Figure 2 ijms-22-10523-f002:**
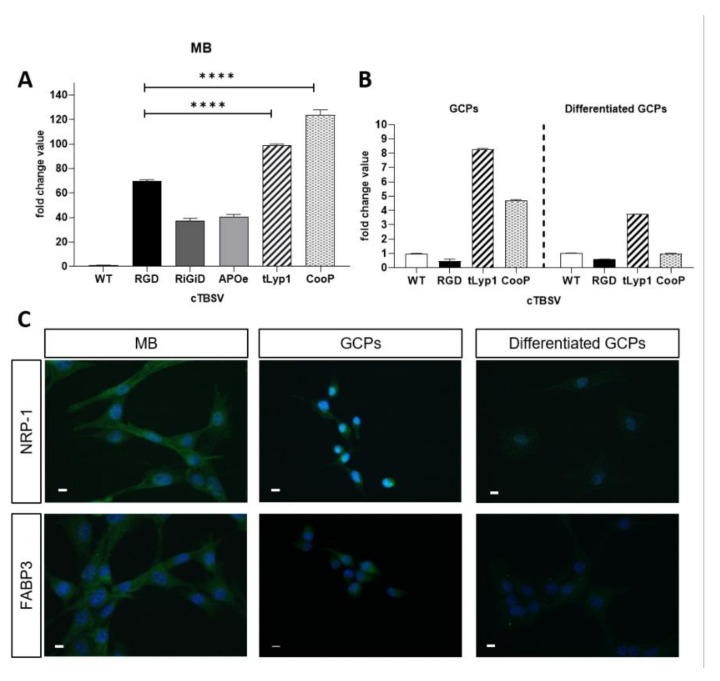
TBSV NPs uptake and receptors immunolocalization**.** Quantification through qPCR of cTBSV internalized by (**A**) Shh-MB primary cells, (**B**) GCPs (left panel) and differentiated GCPs (right panel) using primers specific for a region in the *cp* gene common to all cTBSV NPs. Data are presented as mean ± SD of biological triplicates. Values obtained with TBSV-WT were taken as 1. (**C**) Localization by immunofluorescence of tLyp1 and CooP peptides interacting partners NRP-1 and FABP3, respectively. NRP-1 (top line) and FABP3 (bottom line), in Shh-MB, GCPs and differentiated GCPs. Bars: 10 µm. **** *p* < 0.0001 (Student’s *t*-test).

**Figure 3 ijms-22-10523-f003:**
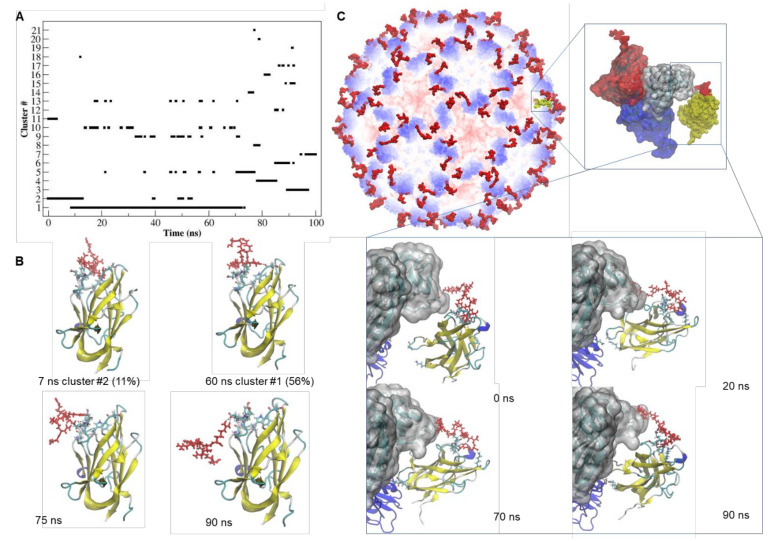
Dynamics and stability of the interaction of tLyp1-NRP-1 and TBSV CPtLyp-NRP-1**. (A)** Number of clusters as a function of simulation time and their occupancy throughout the trajectory of the tLyp1-NRP1 complex. (**B**) Simulation snapshots of the tLyp1-NRP1 complex, taken at selected times of the MD trajectory. The residues of NRP-1 involved in the interaction with tLyp1 are represented by licorice model. tLyp1 is colored in red and depicted in licorice model. The representative structures of the most populated clusters and their percentage of occupancy along the trajectory are also indicated. Black dashed lines: hydrogen bonds. (**C**) Structural model of TBSV-tLyp1 in complex with NRP-1. Zoom views: simulation snapshot of the asymmetric unit of the cTBSV docked with NRP-1 receptor (shown in yellow and by Surface representation model) and simulation snapshots of the CPs of the asymmetric unit docked with NRP-1 (shown by New Cartoon representation model) taken at selected times of the MD trajectory. The tLyp1 peptide is depicted in licorice model and colored in red.

**Figure 4 ijms-22-10523-f004:**
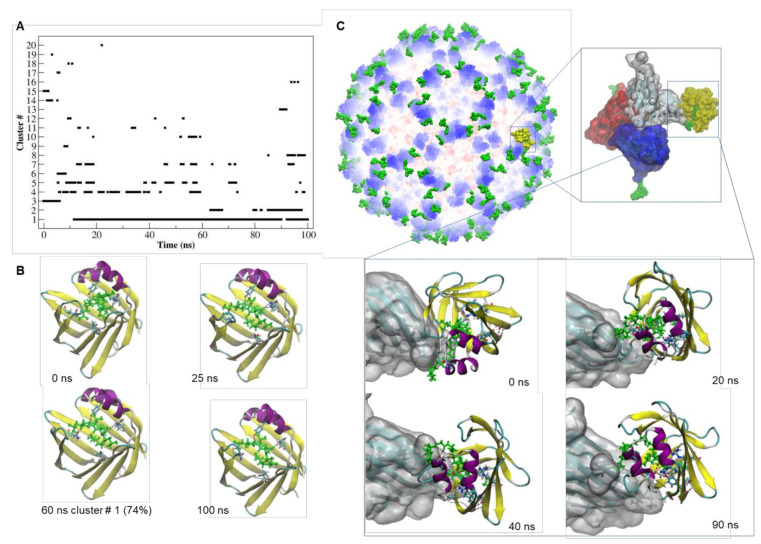
Dynamics and stability of the interaction of CooP-FABP3 and TBSV CPCooP-FABP3. (**A**) Number of clusters as a function of simulation time and their occupancy throughout the trajectory of the CooP-FABP3 complex. (**B**) Simulation snapshots of the CooP-FABP3 complex, taken at selected times of the MD trajectory. The residues of FABP3 involved in the interaction with CooP are represented by licorice model. CooP is colored in green and depicted in licorice model. The representative structures of the most populated clusters and their percentage of occupancy along the trajectory are also indicated. Black dashed lines: hydrogen bonds. (**C**) Structural model of cTBSV in complex with FABP3. Zoom views: simulation snapshot of the asymmetric unit of the cTBSV docked with FABP3 receptor (shown in yellow and by Surface representation model) and simulation snapshots of the CPs of the asymmetric unit docked with FABP3 (shown by New Cartoon representation model) taken at selected times of the MD trajectory. The CooP peptide is depicted in licorice model and colored in green.

**Figure 5 ijms-22-10523-f005:**
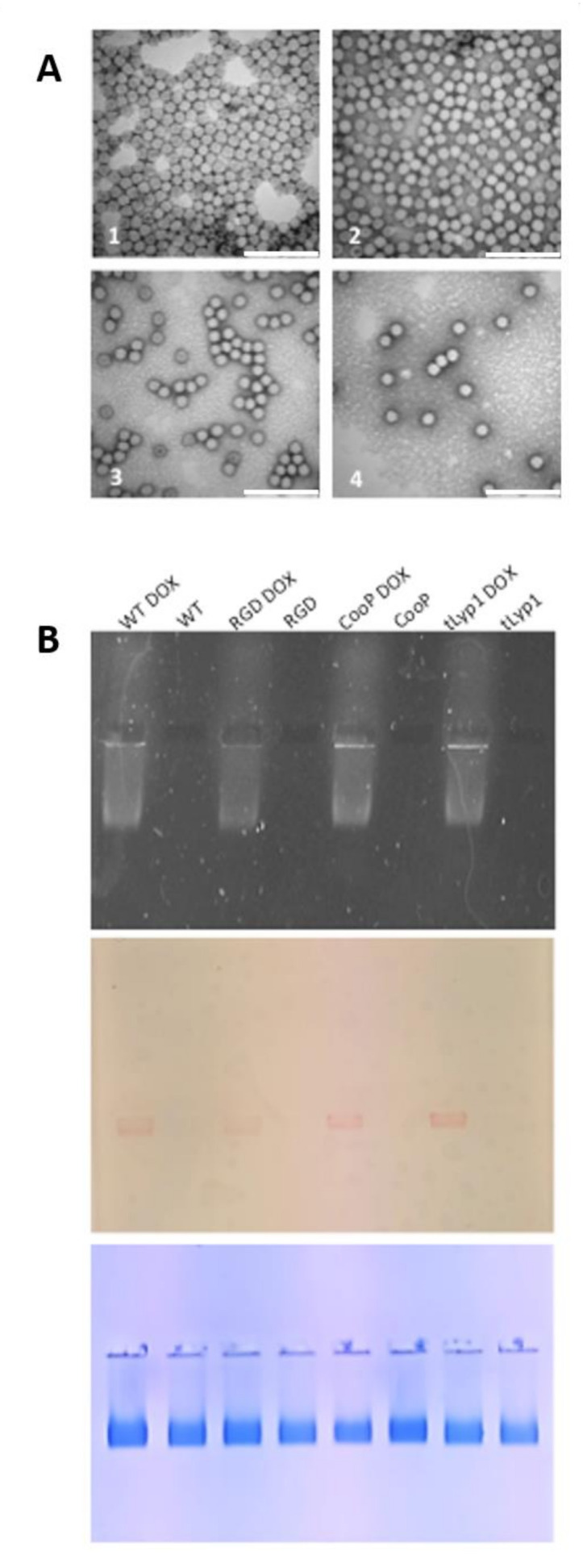
TBSV NPs transmission electron microscopy and DOX loaded TBSV NPs native agarose gel electrophoresis. (**A**) Visualization by negative staining and high-resolution transmission electron microscopy of purified TBSV NPs. (1): TBSV-WT NPs; (2): TBSV-RGD NPs; (3): TBSV-tLyp1 NPs; (4): TBSV-CooP NPs. Bar scale: 200 nm. (**B**) Native agarose gel visualized under UV light (upper panel), and white light before (middle panel) and after Coomassie blue staining (lower panel). Four µg of NPs were loaded in each well. TBSV NPs, loaded or not with DOX, are indicated for each well.

**Figure 6 ijms-22-10523-f006:**
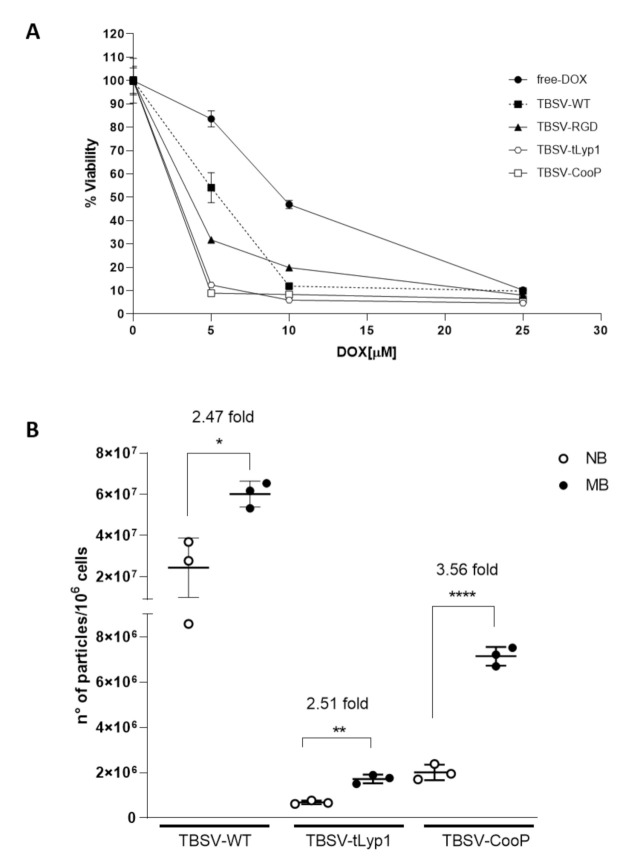
Validation of in vitro and in vivo TBSV targeting to MB. (**A**) Viability of Shh-MB cells after 72 h of incubation with free-DOX or DOX-loaded NPs (TBSV-WT, -RGD, -tLyp1 or -CooP). The viability of the cells treated with the unloaded WT virus was set as 100%. Data are presented as mean ± SD of biological quintuplicates. (**B**) Absolute quantification of TBSV NPs in MB and NB determined by qPCR after IV injection of 200 µg TBSV-CooP, TBSV-tLyp1 or TBSV-WT in *Ptch1^+/−^* mice with clear signs and symptoms of MB. Data are presented as mean ± SD of biological triplicates. * *p* = 0.0171; ** *p* = 0.0010; **** *p* < 0.0001 (Student’s *t*-test). NB—normal brain; MB—Medulloblastoma.

**Table 1 ijms-22-10523-t001:** Characterization of TBSV nanoparticles in DLS/ζ-potential evaluation and DOX loading.

NP	Size * (d.nm) by TEM	Size ^§^ (d.nm) by DLS	PDI	ζ-Potential (mV)	DOX Molecules/Particle ^§^	Ng DOX/ µg Virus	[µM]	EE ^§^ (%)	LC ^§^ (%)
TBSV-WT	32.7 ± 1.1	50.8 ± 18.4	0.429	−5.73	2009 ± 73	130.2 ± 4.7	207	38 ± 6	12 ± 2
TBSV-RGD	32.7 ± 0.9	48.5 ± 14.2	0.489	−5.95	1911 ± 394	136.4 ± 21.2	155	34 ± 1	11 ± 1
TBSV-CooP	32.7 ± 0.8	68.1 ± 17.7	0.588	−3.98	2174 ± 326	151.9 ± 15.9	379	47 ± 5	15 ± 2
TBSV-tLyp1	32.7 ± 0.9	66.5 ± 18.8	0.419	−3.61	2498 ± 749	174.1 ± 32.1	207	54 ± 10	17 ± 3

NP—nanoparticle; PDI—polydispersity index; EE—encapsulation efficiency; LC—loading capacity. ^§^ The data obtained represent the mean value of three measurements ± standard deviation. * The data obtained represent the mean value of 50 measurements ± standard deviation.

## Data Availability

Other datasets analyzed during the study are available from the corresponding authors on reasonable request.

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
