# Peer review of "Tomato Bushy Stunt Virus Nanoparticles as a Platform for Drug Delivery to Shh-Dependent Medulloblastoma"

_ijms, 2021, doi:10.3390/ijms221910523_

Round 1
Reviewer 1 Report
Please see attached file.

Author Response
Reviewer 1
Thanks to the authors for sharing this interesting work. Please consider the comments below that can help add clarity for the broad readership of this journal. Also, your work could benefit from more details in the materials and methods sections, which will help others reproduce this work.
- 2.1 Construction, production and purification of WT and chimeric TBSV NPs
o I cannot find in this section, or in the Materials and Methods, the stops for the production of the nanoparticles, their isolation from the leaves and their purification. Did you use FPLC? Centrifugation? Did you test their endotoxin levels? Was a sterility test performed before administering them to cells or animals?
Some details have been added in the Materials and Methods section to clarify the purification protocol.
As far as sterility and endotoxin evaluation is concerned, in this paper we did not performed these tests. However, this is the purification approach we have commonly used over the years, always with undetectable levels of LPS and bacteria contaminations. Moreover, in the published manuscript (Lico et al., Plant-produced potato virus X chimeric particles displaying an influenza virus-derived peptide activate specific CD8+ T cells in mice. Vaccine. 2009 Aug 13;27(37):5069-76) we presented, although for another plant virus but with similar infection and purification procedures, that the preparation was endotoxin free. In the present paper, cells and mice did not show any adverse effect imputable to any kind of sample contamination.
- 2.2. In vitro validation of cTBSV uptake
o Please elaborate on what the appropriate medium is for the culture of Shh-MB cells. - Figure 2 caption
o Please clarify in the caption that CooP binds to FABP3 and that tLyp1 binds to NRP-1. It can be found in the text of section 2.2, but for someone reading the figure and caption, it is unclear and confusing.
o Also, this caption should include the gene that you performed qPCR on, or refer readers to the supplementary table.
In M&M section we added details about media used for the culture of Shh-MB cells. All other requests have been added in the caption of Figure 2.
- 2.3. Immunolocalization of the interaction partners
o Altogether these results suggested 164 that the specificity of the uptake of TBSV-tLyp1 and TBSV-CooP by MB cells and GCPs 165 may be receptor-mediated
o Is it known in the literature that differentiated GCPs have downregulated FABP3 and
NRP-1? If you have PCR data or a reference, please include it. If the suggestion is that the uptake mechanism is FABP3 and NRP-1, then you have to show that (1) their blockade stops uptake (e.g. treating with antibodies against FABP3 and NRP-1) or that you lack FABP3 and NRP-1 on the cells (PCR showing expression is lower than undifferentiated GCPs and MB). The differentiated cells might just be refractory to uptake in general as well.
Our immunofluorescence experiment shows that MB cells and GCPs express high levels of both NRP-1 and FABP3, that conversely are not detected in differentiated GCPs according with quantitative uptake results showed in Fig. 2B. Nevertheless, the authors agree with the reviewer that the receptor-mediated internalization of the virus cannot be considered the unique uptake via as we discussed also in the text; in fact, also TBSV-WT is internalized, although with a very low efficiency. Future studies will focus to clarify this aspect.
- Table S2
o Why was HADDOCK used for OLA-FABP3, when ClusPro was used for the others?
ClusPro is considered a reliable and useful tool for protein-protein or, as in our case, peptide-protein docking, but is not appropriate for ligand-protein docking. Conversely, HADDOCK is considered an excellent tool for docking chemicals, such as oleic acid (OLA), to macromolecules. This concept is now addressed in the caption of Table S2 of the revised manuscript.
- Figure S1
o It is difficult to read the Cluster # vs Time graphs; can you please include higher resolution versions?
The quality and resolution of the cluster plots of Figure S1 and Figure S2 have been improved in the revised version of the manuscript.
- Supplemental Materials and Methods
o If you are able to upload your model so others can use/adapt it (e.g. put it on GitHub), that’d be great.
Our simulation model cannot be enclosed in a single code. Indeed, our in silico protocol combines different computational approaches (docking, homology modeling, MD simulation, free energy calculations). Some of these have been used directly by on line server (ClusPro and HADDOCK), the others (Modeller, Gromacs, MM/PBSA) require the installation of codes/software on workstation or HPC clusters. However, we partially accepted the Referee’s request by uploading our structural models on the GitHub repository under a project denoted “TBSV-nanoparticles-as-a-platform-for-drug-delivery-to-Shh-dependent-medulloblastoma-PDB-models” (https://github.com/mspodda/TBSV-nanoparticles-as-a-platform-for-drug-delivery-to-Shh-dependent-medulloblastoma-PDB-models). The link is now inserted in the SI of the revised manuscript.
- General note about supplementary tables and figures
o Please add more descriptive captions, adding in your major points, results or conclusions. It’s a bit difficult to jump between the supplemental section and your main text, especially when the captions are little more than titles.
The captions of Figure S1, S2 and of Tables S2, S3 and S4 have been modified in order to make them descriptive. In particular the major results was added at the end of each captions
- Figures 3a and 4a are very blurry and we cannot clearly read the chart.
The quality and resolution of the cluster plots of Figure 3A and Figure 4A have been improved in the revised manuscript.
- 2.5 TBSV-tLyp1 and TBSV-CooP characterization and loading with DOX, Table 1 4.8 Dynamic light scattering and zetapotential analysis
o Please clarify what type of DLS measurements are you reporting. Is this size by intensity, number or volume? While sizes by intensity is independent of refractive index, size by number or volume does require you to know the refractive index of your materials. If you used number or volume, please report the refractive index you used for your nanoparticles. Also, please confirm you are reporting the mean diameter of the peak and not the z-average.
The DLS measurements represent the size by intensity, and the data reported represent the diameter of the peaks and not the z-average. We have added these details in the Material and Methods section.
- Figure 6
o Please indicate which statistical test was performed in the caption (t-test)
We added this information in the revised manuscript
- 4.11 Absolute quantification of c-TBSV in target tissues after intravenous injection
o I recognize the work done on using the Absolute Quantification Method, but the assumptions made (based on your pilot experiment) to support the statement that all nanoparticles went to the brain are still theoretical. Organs like the liver and kidney, which help clear IV injected nanoparticles, should have also been tested to prove experimentally that there is no other uptake in these cells types. This is standard practice for all Absorption, Distribution, Metabolism, and Excretion (ADME) or pharmacokinetic-toxicology studies. If you are unable to do the qPCR analysis of other major organs, please revise your statements that suggest complete uptake with the caveat that it is theoretical and further biodistributions studies are necessary to confirm.
Many thanks for your suggestion, that higlhights a possible difficulty in the comprehension of the text. We have changed our statements to avoid future misunderstandings. In fact, the sentence “The analysis (Fig. 6B) indicated that all NPs were found in the brain” would mean that TBSV-WT, as well as TBSV-tLyp1 and –CooP (all type of our NPs), are able to gain access to the brain, not that all the amount of the injected virus is entirely found in the brain.
- Sex-stratified analysis
o To enable future meta analysis and eliminate potential bias is your work, please disclose the sex of the mice used in your abstract and materials & methods section.
We used symptomatic mice of both sexes (4 females and 5 males). This information has been added in abstract and M&M section
Reviewer 2 Report
Comments: In this manuscript the authors on Tomato Bushy Stunt Virus nanoparticles as a drug delivery vehicle for cancer treatment specially the SHH-dependent subtype. The authors have used a unique method for the synthesis of Virus nanoparticles. These TBSV particles have used for the targeted killing of the MB symptomatic mice model in order to reduce early and late toxicity. Receptors immunolocalization, Molecular Docking and Molecular Dynamics Simulations data also support the experimental results. This paper is well written and I would recommend for the publication after addressing minor issues.
In Table 1, why DOX molecules/ particles, sizes are very big more than 2 mm compare to virus NPs. Explain with proper evidences.
Author Response
In Table 1, why DOX molecules/ particles, sizes are very big more than 2 mm compare to virus NPs. Explain with proper evidences.
To support this Reviewer in better understanding data shown in Table 1, we clarify that the TEM (and DLS) analysis is performed on NPs not loaded with the Doxorubicin and actually the mean diameter reported of 32 nm for all the NPs analyzed is consistent with the data reported in literature for the wt virion (Kumar et al., Tomato bushy stunt virus (TBSV), a versatile platform for polyvalent display of antigenic epitopes and vaccine design. Virology. 2009;388(1):185-190, reporting that “the native capsids of TBSV are ∼320 Å in diameter and are composed of 180 copies of a single 41 kDa CP subunit arranged with T= 3 icosahedral symmetry). Accordingly, we have corrected the reported diameter of the wt virus in the revised text, in the Introduction section. In the TEM image the bar corresponds to 200 nm and the dimensions of the NPs in the photographs are in line with the measurements reported in the table 1. The only data reported about a different and bigger dimension concerns DLS, but this probably highlights the tendency of the NPs to aggregate, and not that virions have a larger diameter, as verified by TEM analysis and specified in the text.